# The Effect of Water Availability on the Carbon Content of Grain and Above- and Belowground Residues in Common and Einkorn Wheat

**DOI:** 10.3390/plants13020181

**Published:** 2024-01-09

**Authors:** Ivana Raimanova, Pavel Svoboda, Michal Moulik, Jana Wollnerova, Jan Haberle

**Affiliations:** Department of Sustainable Arable Land Management and Cropping Systems, Crop Research Institute, 16106 Prague, Czech Republic; svoboda@vurv.cz (P.S.); michal.moulik@vurv.cz (M.M.); wollnerova@vurv.cz (J.W.); haberle@vurv.cz (J.H.)

**Keywords:** chaff, stubble, roots, carbon balance, water availability, harvest index

## Abstract

The carbon (C) fixed by crops, which is exported with harvest and retained as postharvest residues in a field, is important for calculating the C balance. The aim of this study was to determine the effect of water availability on the C content in whole wheat plants. In a three-year field trial, the weights of grain, straw, chaff, stubble, and roots of two cultivars of winter wheat (*Triticum aestivum* L.) and one cultivar of einkorn wheat (*Triticum monococcum* L.) and their carbon contents were determined in water stress, irrigation, and rain-fed control treatments. The water availability, year, and cultivar had a significant influence on the C content in aboveground plant parts, but the effect of water on grain C was weak. The C content decreased with irrigation and increased with drought, but the differences were small (at most, 3.39% in chaff). On average, the C contents of grain, straw, chaff, and roots reached 45.0, 45.7, 42.6, and 34.9%, respectively. The amount of C exported with grain and left on the field in the form of postharvest residues depended on the weight of the total biomass and the ratio of grain to straw and residue. Whole plant C yield reached 8.99, 7.46, and 9.65 t ha^−1^ in rain-fed control, stressed, and irrigated treatments, respectively, and 8.91, 9.45, and 7.47 t ha^−1^ in Artix, Butterfly, and Rumona, respectively. Irrigation significantly increased the C content in grain and straw (but not in chaff, stubble, and roots) in comparison with water shortage conditions. On average, a grain yield of 1 t ha^−1^ corresponded to an average export of 0.447–0.454 t C ha^−1^ in the grain of all cultivars and inputs of 0.721, 0.832, and 2.207 t C ha^−1^ of residue to the soil in the form of straw and postharvest residue in the two cultivars of common wheat and one of einkorn. The results of the study provided reliable data for the calculation of the C balance of wheat under conditions of different water availability.

## 1. Introduction

Crops play a vital role in capturing and storing carbon dioxide from the atmosphere through photosynthesis. Carbon (C) forms almost half of the plant biomass content, and calculation of the C content in crops is important for determining the C balance in agroecosystems and the effort to manage soil quality, sequester C in soil, and mitigate climate change. Plants build C into organic molecules, such as glucose, cellulose, or lignin, which are, in the form of above- and belowground postharvest residues, transformed into soil organic matter by the activity of soil organisms. Plant C content is one of the most important plant traits and is critical to the assessment of the global C cycle and ecological stoichiometry [1]. Attention has been focused on C in recent years following strategies such as the Green Deal and “Farm to Fork”, which are aimed at new ways of sequestering CO_2_ through “carbon farming” and state that farming practices that remove CO_2_ from the atmosphere should be rewarded [2,3,4].

This effort focuses on the adjustment or alteration of crop systems and agronomic measures, especially in terms of biomass input and maintenance of soil organic matter. Postharvest crop residues are the main C source for arable lands when not only yield but also other aboveground biomass (for example, straw in cereals) is removed from a field. The calculation of the benefits of different crop systems and agronomic measures, calculation of the C balance on the field, crop rotation, or farm scale is often based (in addition to the input of C in farmyard manure) on the average C content in plants, estimated amount of postharvest residues, and allometric coefficients of yield to residues used for a wide range of conditions [5,6,7,8]. Common calculations use the harvest index (HI), defined as the ratio of economic yield to aboveground weight, and in the case of cereals, the ratio of harvested grain to total shoot dry matter [9]. Ismail [10] drew attention to some confusion in the methods of assessing the HI, as the aboveground biomass does not always include the total matter of the shoots but only the mechanically harvested part, i.e., without chaff, stubble, and segments of shoots belowground. For example, in wheat, the approximate proportions of grain, straw, and chaff when using a combine harvester are 50, 25, and 25%, respectively [11,12]. Sukhoveeva [7] found that postharvest residues (roots and stubble) of cereal and leguminous crops accounted for 1/4–1/5 of the dry phytomass; the stubble portion was 1/6–1/10 of the phytomass, and the share of roots did not exceed 10%. The portion of grain and chaff was 1/2–1/3 and that of by-products (stems and leaves) was 1/3 for cereals. Chaff sometimes refers not only to the remains of ears and spikelets but also to unharvested grain fragments and lost kernels, dead dry leaves, and finely chopped straw, which further complicates the comparison of data from different sources [13].

Root matter is not normally included in the HI calculation because of the laborious methodology involved and the uncertainty in interpreting the obtained data [14], although exceptions can be found [15]. With data on roots, it is possible to define a “biological yield”, which includes total dry matter production [9]. Poeplau et al. [16] stated that crop roots are the major source of soil organic matter, as root C has a residence time in soils that is two to three times longer than the C of other crop residues and is therefore more effective at maintaining and building soil organic matter [17].

Relatively little is known about the impact of water availability on C content in crops and the consequences of drought or irrigation on C input to the soil in the form of postharvest residues. According to a review by Donald and Hamblin [9], the HI of grain crops is relatively stable, but under dry conditions, the HI is one of the main traits determining grain yield variability, and it is strongly affected by post-anthesis water use [18,19]. Unkovich et al. [20] suggested that the HI can also be used to estimate the C balance, but such a C accounting practice is sensitive to changes in the HI due to climate and soil conditions. This aspect is gaining importance with the onset of climate change, increasing rainfall fluctuation, and water availability [21]. Artificially controlled water availability practices, including drought induction by covering the crop canopy with a rain-out mobile shelter, can model future conditions to some extent [22].

Cultivation of winter wheat, the most important commercial crop with the largest area in the Czech Republic (854 thousand ha in 2022, i.e., 34.4% of arable land) necessarily has an impact on the carbon balance in agriculture [23]. Worldwide, wheat has the largest area of all crops, at over 222 million ha [24]. This study was conducted in an effort to improve the calculation of the C balance in the whole wheat plant under contrasting water availabilities during grain growth.

The aim of this study was to determine the effect of drought and irrigation on the amount (yield) of C in wheat grain, straw, and postharvest residues. The working hypothesis assumed, based on the literature data, that the C content of plants (plant organs) was stable and would not be significantly affected by different water supplies, while the content and proportion of postharvest residues would be affected by different water supplies. In this study, data on C content and the amount of crop residues enabling reliable calculation of the C exported and retained in a field were collected in a three-year field experiment with wheat under contrasting water supplies.

## 2. Materials and Methods

### 2.1. Field Experiment

The field experiment was carried out from 2020 to 2022 in Ruzyně, near Prague, Czech Republic (49°53′29″ N, 15°23′38″ E). The field is situated on fertile deep clay and clay-loam soil formed on loess (Haplic Chernozem soil on loess). The field water capacity (FWC), wilting point, and available water capacity of soil layers down to 90 cm are 34.5–37.2 vol. %, 15.7–16.2 vol. %, and 18.8–21.0 vol. %, respectively. The characteristics were determined repeatedly in the experimental field. The altitude is 340 m, the long-term average temperature is 9.6 °C, and the average annual precipitation is 497.5 mm. Precipitation, temperature, humidity, solar radiation, and wind speed were measured at a station 100 m away from the experimental site (www.vurv.cz/meteostanice, accessed on 7 January 2024). Temperature, precipitation, potential evapotranspiration, and water balance, accumulated from 1st January of the experimental years are shown in Figure 1.

The commercial cultivars of common wheat (*Triticum aestivum* L.) cultivars Artix and Butterfly and einkorn wheat (*Triticum monococcum* L.) cultivar Rumona were studied in this experiment. The cultivars Artix and Butterfly are commonly grown in the Czech Republic, and the einkorn cultivar Rumona is used in ecological farming. Wheat was sown on October 13, 22, and 15 in 2019, 2020, and 2021. Varieties reached maturity in the order of Artix, Butterly, and Rumona, during July to mid-August, depending on weather and water availability. The sowing density was 4.0 million germinating seeds per hectare for common wheat and 3.5 million for einkorn, and the row spacing was 12.5 cm. A total of 100 kg N ha^−1^ was applied, of which 40 kg was applied during the spring regeneration stage at the beginning of tillering (BBCH 20–22), and the remainder was applied at the beginning of stem elongation (BBCH 30–31). The experiment took place on one field divided into three parts, within a 3-year crop rotation of 1. buckwheat, 2. legume–cereal mixture, and 3. experimental wheat, so the experiment was placed on a different plot every year. Standard tillage, deep plough after pre-crop, and plant protection practices were performed.

Three treatments were studied: drought (S), irrigation (IR), and rain-fed control (CON). Drought was induced with the use of a mobile shelter, which was a metal structure with a special transparent plastic polycarbonate roof and walls, moving on rails placed on the trial area at the start of growth. Rainwater was drained by gutters outside the experimental area.

The shelter was used only during rain, from the end of stem elongation onwards so that the content of available water to a depth of 90 cm decreased by crop depletion under 30% of AWC at the start of anthesis (BBCH 60–65) and dropped near the wilting point during grain filling. Drip irrigation was applied from the beginning of anthesis to maintain an available water content above 75% of AWC in the 90 cm soil zone. The water content was manipulated using the calculation of water depletion by evapotranspiration [25], data on winter wheat water depletion in the same field from previous years, and gravimetrical soil moisture determined during growth. In the years 2020–2022, precipitation sums from January 1 to July 15 reached 267 mm, 350 mm, and 303 mm (Figure 1); totals of applied irrigation were 186 mm, 207 mm, and 158 mm, respectively. From the S variant plots, 128 mm, 175 mm, and 167 mm of precipitation were drained away from anthesis in the three experimental years. Natural conditions of the site soil and agrochemical data in the experimental field are described in more detail by Kurešová et al. [26] and Raimanová et al. [27]. This experiment had a cross-sectional design, with blocks of water availability treatments and 2 (S, IR) or 4 (CON) replications. Plots were 5 m × 6 m or 6 m × 6 m in size. Harvest was carried out with a small plot combine harvester.

### 2.2. Plant Sampling

The determination of the C content in whole wheat plants consisted of three steps: (1) determination of dry matter weight of plant parts, (2) determination of C content in these parts, and (3) calculation of C content in grain and residues. At maturity, whole plants were sampled from each plot, and plants were cut at the ground surface. Four samples were collected from an area of 0.25 m^2^ (two from each plot in the S and IR treatments), and the dry mass of ears and straw were determined at 65 °C in a forced-air oven to constant weight. Ears were trashed, and the dry matter of grain was determined. The weights of the residues of ears, chaff, husk, and ear rachis were calculated. The lemma and palea are attached to the seeds in einkorn wheat. To compare the same organs and residues in both wheat species, einkorn grains from several ears and several dozen combine-harvested grains with attached chaff were separated by hand; the ratios of chaff to grain were used for the calculations.

At maturity, five plants were collected with part of the shoot under the soil surface, in four replications (two from each plot in the S and IR treatments) from all treatments. The dry weight of the basal part of the wheat stalk under the soil plus the section of the stalk 10 cm above the soil surface (stubble), the rest of the stalks (straw), and the grain were determined. To describe cases when straw was retained or exported from a field with grain, we distinguished combined straw yield (cut at 10 cm above the soil surface) and total straw (cut at the soil surface). All weights are given for dry matter.

The weight of the straw, chaff, and stubble per plot was calculated from the combine-harvested grain yield, using dry weight rates of grain to the residues determined for whole plants. For the S and IR treatments, average values of two samples from a plot were used.

The roots of wheat were sampled at the end of the grain-filling period (BBCH 87–90). The soil was sampled with a hand sampler with a diameter of 36 mm, in 10 cm increments, in two replicates down to the depth of the last visible roots, to at least a 1 m depth. The average weight of roots taken on a row and within rows represented one replication. The roots were separated with water in sieves and cleaned, and their dry weights and lengths were determined. The methods of root sampling, separating, and cleaning were previously described by Svoboda et al. [28,29]. Pooled samples of roots taken on and within rows were analyzed for C content.

The C content in the dry plant material was determined for four replications using an elemental analyzer EA Vario PYRO cube (Elementar, Langenselbold, Hesse, Germany). Plant material was homogenized into a fine powder in an MM301 ball mill (Retsch, Haan, Germany). The amount of C in grain, straw, and postharvest residues was calculated from the dry matter of the plant parts and the average C content of grain, straw, chaff, and roots.

### 2.3. Statistical Analysis

Statistical evaluation was performed using the STATISTICA 14 program (StatSoft, Inc., Tulsa, OK, USA). The effects of treatments, years, and cultivars were, after confirming the normal distribution of data, analyzed with a three-way analysis of variance (ANOVA), and the differences among means were evaluated with Tukey’s honestly significant difference (HSD) test (at *p* < 0.05).

## 3. Results

### 3.1. Effect of Water Supply on Grain, Straw, and Postharvest Residues

The analysis showed a strong significant influence of all factors, i.e., water supply, year of cultivation, and wheat cultivar, on grain yield (Table 1). Water stress (S) reduced grain yield (*p* < 0.001) in comparison with the control (CON) and irrigation (IR) groups, on average, by 1.86 and 2.26 t ha^−1^, respectively. The difference between C and IR treatments was not significant (on average, 0.41 t ha^−1^). The grain yield was more than twice as high in common wheat cultivars as in the einkorn wheat, Rumona. The difference between Artix and Butterfly was insignificant at only 0.26 t ha^−1^. Significantly lower yields were attained in 2022 compared with the other years.

Similarly, all factors had a significant effect on straw yield. Irrigation increased straw yield by 0.94 and 1.96 t ha^−1^ in comparison with the CON and S treatments, respectively. Einkorn wheat reached a significantly greater straw yield (8.92 t ha^−1^) than common wheat (7.19 and 8.03 t ha^−1^, respectively). This resulted in different ratios of grain/straw (cut at ground) of 1.06, 0.92, and 0.33 in Artix, Butterfly, and Rumona, respectively. In the case of straw cut at a height of 10 cm (the combined harvest), the rates were 1.21, 1.06, and 0.37, respectively. The rates were reduced by water shortage in comparison with CON and IR treatments.

The effect of water treatments on chaff weight corresponded to the effect on grain yield; water stress reduced the weight, and irrigation increased it. Butterfly showed a significantly higher chaff weight that corresponded to a different ear structure in comparison with Artix. Chaff weights reached, on average, 19.5, 26.9, and 49.5% of the grain yield in Artix, Butterfly, and Rumona, respectively. The corresponding rates of stubble weight (part of stalk under the ground + 10 cm straw length) to grain were 24.6, 28.6, and 81.8%, respectively. The effects of year, treatment, and cultivar on stubble weight were significant (Table 1). The response of the roots to different water treatments was not significant.

### 3.2. Effect of Water Supply, Year, and Cultivar on Carbon Content in Grain, Straw, and Postharvest Residues

The C content in grain and straw over three years varied within a narrow range, with averages of 45.05% (±0.89%) and 45.66% (±1.09%), respectively (Table 2). The effects of all studied factors on the C content were significant for straw and grain (*p* < 0.001), except for the influence of the water treatments on grain C (*p* = 0.053). The C content of grain in 2020 (the year with lower yields) was 0.74 and 1.03% higher than in 2021 and 2022, respectively. Similarly, the C content of straw was significantly higher in 2020, by 0.25 and 1.26%, respectively. The higher C content corresponded to higher temperatures in 2020 but not to precipitation or water balance (Figure 1).

A higher C content was observed in straw from the CON and S treatments compared to IR plants; the average difference between contrasting treatments S and IR was 1.39%. This effect was apparent in all experimental years. In grain, the difference was not significant (*p* = 0.053).

The average C content in straw was 0.61% higher than the C content in grain. The C content of grain was significantly higher in einkorn wheat than in common wheat; for straw, the difference was significant only in Butterfly.

The effect of water supply on the C content of chaff was similar but stronger than the effect on grain and straw. Irrigation significantly reduced the C content by 3.39% and 1.39%, on average, in comparison with S and CON plants (Table 2). In addition, the effect of the cultivar was similar to the effect on grain C content, i.e., there were higher values for einkorn wheat than common wheat and a higher C content for Butterfly than Artix, but the differences were not significant.

The C content of chaff (42.48 ± 1.45%) was lower than that of grain and straw (45.05 ± 0.89 and 45.65 ± 1.05%, respectively). The C content of roots was affected by year but not by water availability or cultivar. It was about 10% lower than that of the aboveground parts and the data showed higher variability than observed in shoots.

### 3.3. The Amount of Carbon in Grain, Straw, and Postharvest Residues

The factors studied had highly significant effects on the amounts of carbon in grain, straw, and postharvest residues (Table 3). Due to the low variability in C content among treatments, years, and cultivars, the amount of C in plant parts depended on the effect of the factors on the weight of plant parts. Hence, water shortage (S) significantly reduced the amount of C in grain by 1.00 t ha^−1^, on average, in comparison with the IR treatment and by 0.17 t ha^−1^ compared to the CON treatment. The impact of the water supply was less pronounced in straw and residues.

Whole plant C yields reached 8.99, 7.46, and 9.65 t ha^−1^ in rain-fed control, stressed, and irrigated treatments, respectively. Irrigation increased the whole plant C yield by 0.66 and 2.19 t ha^−1^, on average, compared to the CON and S treatments, respectively. The grain C amounts in common wheat cultivars were 1.99 and 2.08 t ha^−1^ higher than in einkorn wheat, which are in agreement with the higher grain yields in common wheat cultivars. A lower grain C yield was not fully compensated for by a higher C amount in the straw of einkorn Rumona (higher by 0.79 and 0.47 t ha^−1^ in comparison with Artix and Butterfly); C levels in stubble and chaff were similar in the three cultivars. As a result, the whole plant C yield was significantly lower in einkorn wheat Rumona (7.74 t ha^−1^, on average) than in Artix and Butterfly (8.91 and 9.45 t ha^−1^, respectively).

The amounts of carbon retained in postharvest residues (chaff, stubble, and roots) and combine-harvested straw (cut at 10 cm) averaged over years and cultivars reached 6.11, 5.30, and 6.47 t ha^−1^ in CON, S, and IR treatments, respectively (Figure 2). The average amount of C in residues and straw was relatively similar among the cultivars (5.44, 6.17, and 6.29 t ha^−1^ in Artix, Butterfly, and Rumona, respectively). When straw was exported, the corresponding amounts of C retained in the field reached 2.61, 3.00, and 2.82 t ha^−1^.

The average (standard) HI of C (grain C/grain + harvester combine straw C) ranged from 0.426 (S) to 0.479 (CON). The biological HI of C, i.e., the rate of C fixed in and exported from the system with wheat grain to whole plant C yield, represented 0.385, 0.350, and 0.174 of the whole plant C at maturity, on average, in Artix, Butterfly, and Rumona, respectively. Stress reduced the proportion of exported C from 0.385 and 0.350 in the CON and IR treatments to 0.282 in the S treatment.

The biological HI of C in common wheat was twice that of einkorn. In other words, the amount of C in residues per 1 t of grain was 0.72, 0.84, and 2.16 t ha^−1^, on average, in Artix, Butterfly, and Rumona, respectively, when straw was retained in the field and 0.34, 0.41, and 0.97 t ha^−1^, respectively, when straw was exported.

## 4. Discussion

The results of this three-year study confirmed there is a similar, relatively stable C content observed in aboveground organs of wheat. The effects of the studied factors on C content were mostly significant, but the absolute differences among treatments or cultivars were mostly in the range of tenths or one to two percent. The average values of grains and straw were 45.05% (±0.81%) and 45.66% (±1.09%), which are near to the value of 45% used for the calculation of C content in plant dry mass [30,31]; a general value of 50% is also used [32]. Our data are near the average C content in reproductive organs (45.0%) or stems (47.9%) on global scales [1]. For leaves and roots, the average C contents calculated in this study were 46.9% and 45.6%, respectively. Our data for root C was significantly lower than the published one [33,34].

The difference in C content between grain and straw seems low considering their different chemical composition, but cellulose, the main compound of straw, has a C content similar to non-structural carbohydrates (sugar, starch) found in grains, at about 44%. A stronger effect may involve lignin, with a C content of 63–66% [35,36], which might have contributed to the slightly higher C content of straw. Previous studies show a higher lignin content in straw (11–26%) than in grain and wheat bran [37,38,39]. Among the impact of other factors, drought may affect root morphological and physiological traits that modify the cellulose and lignin contents and the decomposability of plant material [40].

The effects of cultivar and water availability on the C content of grain and straw were significant but the differences were small. The differences may be related to the different compositions and the mineral content in the grains of common and einkorn wheat. For example, Brandolini et al. [41] showed that the protein, ash, and pigment values of einkorn grain (meal flour) significantly exceeded those of the *Triticum aestivum* controls, but according to Biel et al. [42], the content of ash was similar in this species.

The C content of chaff was lower than that of grain and straw, by 2.48% and 3.09%, respectively. The ear and constituents of the husk (glumes, lemmas, paleas) and ear rachis are formed earlier during ear development (before differentiation of the water supply in the experiment), while the main grain mass was formed after flowering, during grain filling, and under contrasting water availability. The interpretation of the impacts of various factors is not easy, as C and other elements (especially N) in grain are reutilized and redistributed to grain from reserves in other organs, leaves, roots, and (inter)nodes during grain filling, which may affect chaff composition. This effect may be related to a rather strong difference in chaff C content between irrigated (41.06%) and stressed wheat (44.92%). The effects of various factors on the content of other elements that form ash residues after burning (especially K, Ca, P, and Si) may also affect the proportion of C in plant organs.

The C content of roots varied the most from all analyzed plant parts and was significantly lower than in the aboveground parts (on average 34.91%). For the calculation of C input in soil, a lower C content is used for roots (or belowground material) than aerial parts; for example, Clivot et al. [43] applied a coefficient of 40%. This is surprising, as roots contain more lignin, which has a high content of C [1,35]. The probable reason for the discrepancy is the small mineral particles adhering to the roots of plants, which are difficult to remove during root separation with water [44]. The incombustible mineral particles (ash) reduce the proportion of organic matter and C in the biomass. Confirmation requires incineration of the root material, but for the purposes of this study, the essential factor was the C content in the dry root biomass. The weight of the roots could be corrected (reduced) by the weight of mineral particles if we could distinguish them from ash coming from the uptake of nutrients.

As the C content of grain, straw, and postharvest residues was relatively stable, the main impact on total C yield was the variability in the grain yield and the rate of grain to straw and residues caused by water availability, year conditions, and cultivar. The use of average values of the amount of postharvest residues of a given crop, regardless of environmental conditions, can thus lead to a significant distortion of the carbon balance in the agroecosystem [45]. For example, Stella et al. [46] predicted the contribution of crop residues to soil organic carbon conservation to the year 2050 with zero, one-third, and 100% removal of cereal residues from a field. As expected, water shortage significantly reduced the grain yield and straw in comparison with rain-fed and irrigated wheat. The greatest differences, over 100%, were among the grain yield of common wheat cultivars and einkorn wheat, but the straw yield was similar in both wheat species.

The contribution of chaff and stubble to postharvest residues is often neglected, although it is important for C balances [43]. Chaff represented 9.7–12.2% of total aboveground biomass, on average, in the examined cultivars, which falls within the range of ratios from 8 to 18% given by Shand et al. [47] and Kernan et al. [48]. The chaff and stubble weights totaled 3.73 t ha^−1^, on average in the two common wheat cultivars, which is half of the weight of the grain (7.50 t ha^−1^). The rate of chaff/grain in einkorn wheat (0.496) was twice as high as that of common wheat (0.196 and 0.269), which is in agreement with the progress of the HI during cereal breeding [9,49]. Losses of residues in the form of wheat grains during combine harvesting were not monitored in this experiment, but if considered, they would slightly increase the proportion of residues.

Roots contributed 3.32 t ha^−1^, on average, representing 16.51% of the total plant biomass, but the results were rather variable and inconsistent. Irrigation increased this insignificantly and water stress decreased root weight, contrary to the commonly observed adaptation of plants to drought [50]. The roots in the surface layer 0–10 cm, which represented about half of the total root biomass, could respond to the positive effect of irrigation on maintaining living tillers forming their own roots. Studies on the root biomass and root/shoot (R/S) rate of wheat also vary over a great range, depending on soil, climate, and agronomic conditions, as well as species and cultivars [31,51]. Pausch and Kuzyakov [52] presented 0.15 as the average ratio of crop root mass to aboveground biomass; Chirinda et al. [33] published a significantly higher proportion of roots. The R/S value is expected to express the efficiency of the root system and to be a useful breeding target; larger roots or R/S ratios are expected in old landraces or modern wheat ancestors in comparison with current cultivars, but the published studies are not fully conclusive, as shown by the data from the presented experiment [50,53]. This could be, in the future, a genotype characteristic significant in the context of soil C sequestration.

Irrigation increased the C content in the whole plant (grain, straw, and residues) by 2.17 or 0.52 t ha^−1^ in comparison with stressed and rain-fed plants, resulting in increased amounts of C exported from the system in grain and straw by 1.77 and 0.53 t ha^−1^, respectively. The C content retained in residues (without straw) was also higher. This shows that C was fixed and retained in response to different water availability (modeling dry and wet years), and that evaluation of the C balance may not be simple.

In parallel with the commonly used HI based on grain and straw yield, we calculated the carbon HI. As grain is always exported from the field, the carbon harvest index enables the calculation of the C content in residues left after harvest. Thus, the average rates of grain C/C in grain plus combine-harvested straw were 0.55, 0.51, and 0.28, on average, in Artix, Butterfly, and Rumona, respectively. For the rates of grain C/total C in whole plants (biological carbon HI), the index decreased to 0.38, 0.35, and 0.17, respectively. The main effect on the proportion of C exported/C retained in a field had management of straw, which accounted for 2.83–3.48 t ha^−1^ of C. Retaining straw in the field (usually by incorporation in soil) strongly improves the C balance and contributes to sustaining the organic matter content of the soil.

The roots represent an uncertainty in our calculation due to the variability in the C content and root mass determination. The root dry weight represented 16.5% of the total wheat crop biomass when averaged over all years. Due to a lower C content of roots in comparison with aboveground parts, this resulted in 13.1% of total plant C content, on average. The possible impact of inevitable methodological error (loss of material during the separation of the roots from the soil by washing, contamination of the roots with mineral grains) on the calculations of total C residues may not be so strong when we take into account the small proportion of carbon in the biomass of the roots. Roots represent the least certain part of the description of the C cycle in the soil and in ecosystems, because C is lost during growth to the soil in the form of exudates and mechanically abraded surface tissues and due to the decay of some roots during growth [52]. According to a review of previous studies, the net C deposited in the roots of grasses and dicotyledonous crops represents 16 and 10% of total assimilated C, respectively, and it is an important aspect of soil C sequestration [54]. Clivot et al. [43] calculated plant C extra-root material (C from root turnover and root exudates) as 65% of root C [31]. The losses of root material were documented by a significant decrease in root weight between flowering and maturity in the presented experiment (not shown). Still, the laborious determination of underground parts provides useful data, as roots contribute to C sequestration more than easily decomposing organic materials inputted to soil during growth [16,17]. Furthermore, according to Remus and Augustin [55], there is a strong correlation between root growth and the C content stored in the soil; thus, data on root biomass are important.

## 5. Conclusions

The results of the study showed the main factors determining the amount and proportion of (assimilated) C exported in the form of grain and retained in the form of straw and postharvest residues. The results did not confirm the working hypothesis that the C contents in wheat plants (except for grain) were not significantly modified by water availability. However, the effects of year, water, and wheat cultivars and species on C content were of minor importance, while the effects of the factors on grain yield and the ratio of grain to straw and postharvest residues were more important. Including the weight of chaff, stubble, and roots enabled us to calculate a more realistic amount of C left in the system in the form of postharvest residues. The impact of drought and irrigation, imitating possible extreme events, significantly affected the amount of C exported and left in a field after harvest. The influence of these factors must be assessed in the context of carbon transformation processes in the agroecosystem and from the point of view of the dynamics of organic soil matter. Further research should also focus on collecting data on C yield and the proportion of C in residues of other important crops, especially spring barley, oilseed rape, grain silage and maize, and potatoes.

## Figures and Tables

**Figure 1 plants-13-00181-f001:**
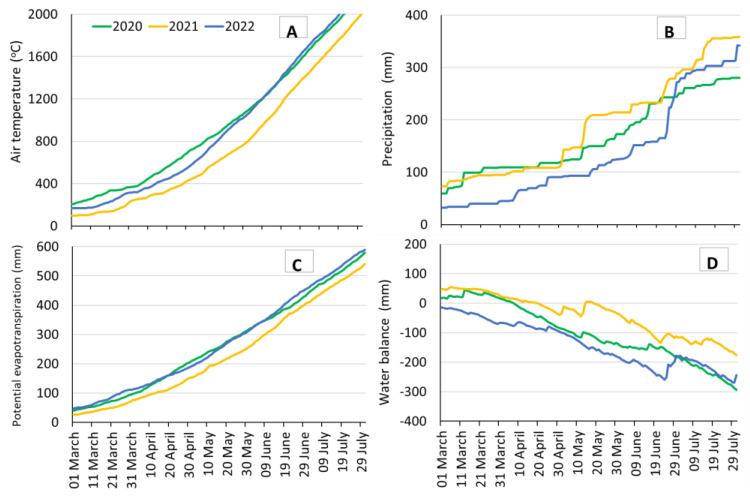
Air temperature at 2 m (**A**), precipitation (**B**), potential evapotranspiration (**C**), and water balance (**D**) accumulated from 1 January at the experimental site Ruzyně.

**Figure 2 plants-13-00181-f002:**
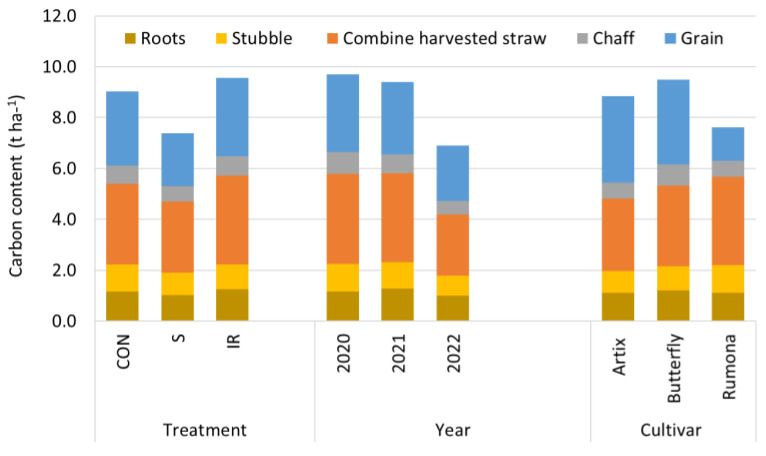
Average values of carbon content of wheat plant components (grain, chaff, combine-harvested straw, stubble, and roots).

**Table 1 plants-13-00181-t001:** The effect of water availability treatment (water stress (S), control (CON), and irrigation (IR)), year of cultivation, and two cultivars of common wheat (Artix, Butterfly) and einkorn wheat (Rumona), on yield of grain and straw and postharvest residues. Average ± std. deviation (in t ha^−1^).

Factor	Grain	Straw ^1^	Chaff	Combine Straw ^2^	Stubble	Roots
	t ha^−1^					
Treatment						
Control	6.45 ± 2.68 ^b^	8.06 ± 1.82 ^b^	1.72 ± 0.53 ^a^	6.87 ± 1.50 ^b^	2.32 ± 0.56 ^a^	3.29 ± 0.49
Stress	4.60 ± 2.14 ^c^	7.06 ± 1.99 ^c^	1.35 ± 0.47 ^b^	6.21 ± 1.88 ^b^	1.89 ± 0.58 ^b^	2.97 ± 1.19
Irrigation	6.86 ± 2.48 ^a^	9.02 ± 2.11 ^a^	1.85 ± 0.54 ^a^	8.03 ± 1.99 ^a^	2.16 ± 0.47 ^a^	3.69 ± 0.86
*p*	<0.001	<0.001	<0.001	<0.001	<0.001	ns
Year						
2020	6.67 ± 3.09 ^a^	8.94 ± 1.09 ^a^	1.98 ± 0.60 ^a^	7.84 ± 0.90 ^a^	2.33 ± 0.44 ^a^	3.18 ± 1.01 ^b^
2021	6.34 ± 2.51 ^a^	8.83 ± 1.90 ^a^	1.70 ± 0.26 ^b^	7.74 ± 1.91 ^a^	2.28 ± 0.42 ^a^	3.97 ± 0.70 ^a^
2022	4.90 ± 2.04 ^b^	6.36 ± 2.09 ^b^	1.23 ± 0.53 ^c^	5.53 ± 1.91 ^b^	1.76 ± 0.61 ^b^	2.80 ± 0.67 ^b^
*p*	<0.001	<0.001	<0.001	<0.001	<0.001	<0.05
Cultivar						
Artix	7.63 ± 1.60 ^a^	7.19 ± 1.86 ^b^	1.49 ± 0.54 ^b^	6.33 ± 1.67 ^b^	1.88 ± 0.43 ^c^	3.29 ± 1.10
Butterfly	7.37 ± 1.94 ^a^	8.03 ± 2.01 ^b^	1.98 ± 0.60 ^a^	6.93 ± 1.76 ^b^	2.11 ± 0.69 ^b^	3.56 ± 0.73
Rumona	2.91 ± 0.70 ^b^	8.92 ± 2.15 ^a^	1.44 ± 0.38 ^b^	7.84 ± 2.10 ^a^	2.38 ± 0.40 ^a^	3.10 ± 0.89
*p*	<0.001	<0.001	<0.001	<0.001	<0.001	ns

Note: The means of the given factor (Year, Treatment, Cultivar) with the same letter do not differ significantly at *p* ≤ 0.05. ^1^ Stalks cut at the ground surface. ^2^ Combine harvest, stalks cut 10 cm above soil.

**Table 2 plants-13-00181-t002:** Effect of water treatments (water stress (S), control (CON), and irrigation (IR)), year, and cultivar (Artix, Butterfly, Rumona) on the carbon content of grain, straw, chaff, and roots. Average ± std. deviation (in %).

Factor	Grain	Straw	Chaff	Roots
	%			
Treatment				
Control	45.15 ± 0.70	45.74 ± 0.81 ^b^	42.19 ± 0.58 ^b^	35.33 ± 4.74
Stress	45.15 ± 0.77	46.31 ± 1.12 ^a^	44.45 ± 1.26 ^a^	35.52 ± 4.74
Irrigation	44.84 ± 0.95	44.92 ± 0.89 ^c^	41.06 ± 0.85 ^c^	33.88 ± 4.74
*p*	0.053	<0.001	<0.01	ns
Year				
2020	45.64 ± 0.54 ^a^	46.16 ± 1.36 ^a^	42.74 ± 2.2 ^a^	36.97 ± 5.27 ^a^
2021	44.90 ± 0.89 ^b^	45.91 ± 0.62 ^b^	42.61 ± 1.49 ^a^	32.01 ± 3.69 ^b^
2022	44.61 ± 0.61 ^b^	44.90 ± 0.74 ^c^	42.34 ± 1.48 ^a^	35.76 ± 3.77 ^a^
*p*	<0.001	<0.001	ns	<0.01
Cultivar				
Artix	44.63 ± 0.62 ^c^	45.86 ± 0.59 ^a^	41.95 ± 1.31 ^b^	34.63 ± 4.58
Butterfly	45.00 ± 1.01 ^b^	45.18 ± 0.92 ^b^	42.71 ± 1.77 ^ab^	34.02 ± 3.30
Rumona	45.52 ± 0.46 ^a^	45.93 ± 1.47 ^a^	43.03 ± 1.95 ^a^	36.08 ± 5.91
*p*	<0.001	<0.001	0.047	ns

Note: The means of the given factor (Year, Treatment, Cultivar) with the same letter do not differ significantly at *p* ≤ 0.05. ns—not significant.

**Table 3 plants-13-00181-t003:** The effect of water availability treatment (water stress (S), control (CON), and irrigation (IR)), year of cultivation, and two cultivars of common wheat (Artix, Butterfly) and einkorn wheat (Rumona) on the amount of carbon in grain, straw, and postharvest residues. Average ± std. deviation (in t ha^−1^).

Factor	Grain	Straw ^1^	Chaff	Combine Straw ^2^	Stubble	Roots
	t ha^−1^					
Treatment						
Control	2.91 ± 1.22 ^a^	3.69 ± 0.79 ^ab^	0.73 ± 0.21 ^a^	3.17 ± 0.70 ^ab^	1.06 ± 0.25 ^a^	1.15 ± 0.14
Stress	2.08 ± 0.97 ^b^	3.28 ± 0.99 ^b^	0.60 ± 0.22 ^b^	2.80 ± 0.87 ^b^	0.87 ± 0.27 ^b^	1.03 ± 0.35
Irrigation	3.08 ± 1.16 ^a^	4.05 ± 0.93 ^a^	0.76 ± 0.22 ^a^	3.50 ± 0.85 ^a^	0.97 ± 0.20 ^ab^	1.24 ± 0.33
*p*	<0.001	<0.001	<0.001	<0.001	<0.001	ns
Year						
2020	3.05 ± 1.40 ^a^	4.13 ± 0.44 ^a^	0.85 ± 0.24 ^a^	3.55 ± 0.39 ^a^	1.08 ± 0.19 ^a^	1.17 ± 0.38
2021	2.84 ± 1.09 ^b^	4.05 ± 0.83 ^a^	0.73 ± 0.11 ^b^	3.51 ± 0.77 ^a^	1.05 ± 0.20 ^a^	1.27 ± 0.22
2022	2.19 ± 0.89 ^c^	2.85 ± 0.90 ^b^	0.52 ± 0.18 ^c^	2.41 ± 0.77 ^b^	0.79 ± 0.27 ^b^	0.99 ± 0.21
*p*	<0.001	<0.001	<0.001	<0.001	<0.001	ns
Cultivar						
Artix	3.41 ± 0.73 ^a^	3.30 ± 0.87 ^b^	0.63 ± 0.23 ^b^	2.83 ± 0.77 ^b^	0.86 ± 0.20 ^b^	1.11 ± 0.32
Butterfly	3.32 ± 0.89 ^a^	3.62 ± 0.89 ^b^	0.85 ± 0.24 ^a^	3.16 ± 0.77 ^ab^	0.95 ± 0.32 ^b^	1.20 ± 0.23
Rumona	1.32 ± 0.33 ^b^	4.09 ± 0.95 ^a^	0.62 ± 0.11 ^b^	3.48 ± 0.90 ^a^	1.09 ± 0.18 ^a^	1.10 ± 0.35
*p*	<0.001	<0.001	<0.001	<0.001	<0.001	ns

Note: The means of the given factor (Year, Treatment, Cultivar) with the same letter do not differ significantly at *p* ≤ 0.05. ^1^ Stalks cut at the ground surface. ^2^ Combine harvest, stalks cut 10 cm above soil.

## Data Availability

Data and scripts generated and/or analyzed during this study are available from the corresponding author upon request.

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
