# Peer review of "The Effect of Water Availability on the Carbon Content of Grain and Above- and Belowground Residues in Common and Einkorn Wheat"

_plants, 2024, doi:10.3390/plants13020181_

Round 1

Reviewer 1 Report

Comments and Suggestions for Authors

The paper has major scientific merits. However, I have pointed out below the shortcomings only so that the paper meets the scientific rigor required for the Plants journal. 

Comments on the Quality of English Language

Generally, English is good. However, rigorous review among authors should improve the language.

Reviewer 2 Report

Comments and Suggestions for Authors

Dear authors,

I'm very satisfied with your manuscript, and presented results. My opinion and recommendation is that the manuscript must be published. However, after carefully reading the manuscript, I discovered some elements that needed to be improved. All comments are visible in the attached pdf file.

Author Response

Thank you for your time you devoted to careful reading the manuscript and capturing errors; your relevant comments helped to improve our article

The usual order in the manuscript is as follows:

- material and methods, - results,- discussion

Please reorder the chapters

JH: We agree, that order also seemed unusual to us, we would prefer standard one, but we followed the order shown in Template for Plants. At last, we decided to change order as most articles in Plants have standard one, too

Same latter is for Carbon (C) and for control (C). I suggest to change "control" in other letter or letters, e.g. "con" It is necessary to go through the entire text.

JH: It is true, thank you for suggestion, we changed C(ontrol) to CON

Between letter t and ha need to be empty space - without dot (t ha-1). Should be uniform throughout the text

JH: Again, it seems that this is not unified in Plants, we did not find an exact instruction in the Instruction or in the Template, it is true, in the last articles of Plants the units with a space prevail. So, we modified the form of units, accordingly

delete space, italic

JH: Thank you for your careful review and the notice, the errors, typos were corrected

Sowing of experimented cultivars were in monoculture or in crop rotation, and as stationary field experiment or...? Which kind of agritechnics were used (soil tillage, fertilization...). Please explain in short and add missing data.

JH: Based on your comment, we have briefly added the required data. (Since the experiment, site and experimental field conditions was already described in the previous articles, we wanted to partially avoid repeating the text and increasing “plagiarism %”).

Drying up what level? - air dry or 65 °C or?

xJH: Dry matter was determined as a standard for all parts of wheat plants at a temperature of 65 oC. The description has been added.

Some references are old (almost 50 years). My suggestion is to find newest instead old one - of course if it is possible. Reference number: 14, 18, 19, 26, 35.

JH: It is true, but the quotes seemed appropriate for the specific problem. We have replaced two references.

Reviewer 3 Report

Comments and Suggestions for Authors

This is the revision of the manuscript number plants-2797942The effect of water availability on the carbon content of above- 2 and belowground residues in common and einkorn wheat” proposed by Raimanova and colleagues for consideration for publication in Plants. The manuscript determines the effect of drought and irrigation on the content of C in wheat grain, straw, and postharvest residues. For this, a 3-year field study has been carried out with different varieties of wheat. Therefore, it is a manuscript with great interest and which contains a lot of information, which could be a valuable scientific contribution to the knowledge about the effects of climatic changes on C balance in the wheat (a crucial crop for agricultural production). I have read the manuscript and find it of deep and wide interest for the scope of Plants and its readers. Thus, for all those mentioned above reasons I support its publication in this prestigious journal after minor revision.

Minor comments

Introduction:

Authors should make a greater effort to highlight the novelty and importance of their research

Materials and methods:

Figure 1 should include all meteorological data (crop cycle) for the study years. In its current state it only shows data from March to July. However, the crop has been sown in October.

In the subsection 4.1. Field Experiment the authors could show water consumption

Results:

The authors should review the statistical study, as some significant differences in the results shown are striking.

Some examples:

Table 1 effects of treatment on grain C: 6.45±2.68b and Ir: 6.86±2.48a

Idem Table 1 effects of treatment of straw C: 8.06±1.82b and S: 7.06±1.99c

Table 2. Effect of cultivar on carbon content of root a p-value of 0.01 appears when it should be non-significant.

Please review the statistical study thoroughly, in order to make the manuscript more rigorous, and take this aspect into account in case the discussion of the results is affected.

Author Response

Thank you for carefully reading the manuscript, capturing unpleasant mistakes and relevant comments that helped us to improve the quality of the article.

Minor comments

Authors should make a greater effort to highlight the novelty and importance of their research

JH: We have tried to improve Introduction in this regard but we have to admit that it is not easy to change the finished text

Materials and methods:

Figure 1 should include all meteorological data (crop cycle) for the study years. In its current state it only shows data from March to July. However, the crop has been sown in October.

JH: We understand your comment, of course it is possible if it would be your essential requirement. Under the given climatic conditions, sowing is postponed to the middle of October (due to the risk of pest infestation and virus infection), wheat plants overwinter in the 0-2 tillers stage and re-growth, plant regeneration, starts in March, so almost all the C in the plant is assimilated in the next period. The beginning of summation of temperatures and precipitation is from January 1, although we only present data from March 1 for better clarity.

In the subsection 4.1. Field Experiment the authors could show water consumption

JH: The amount of water added by an irrigation and drained away with shelter (treatment S) was added to show the difference in the input of water during growth. The presentation of water consumption would demand more wider explanation and justification. It is possible under site soil-climate conditions, as there is no percolation during growth (maybe once in 10 years) and the balance approach seems reliable (not during winter)

Results:

The authors should review the statistical study, as some significant differences in the results shown are striking. Some examples:

Table 1 effects of treatment on grain C: 6.45±2.68b and Ir: 6.86±2.48a

Idem Table 1 effects of treatment of straw C: 8.06±1.82b and S: 7.06±1.99c

JH: The input data, ANOVA and Tukey test results were checked and are correct. In the first case (the grain yield), the difference between the value of C and Ir was significant at p=0.0156. In the second case (straw yield) p=0.022.

Table 2. Effect of cultivar on carbon content of root a p-value of 0.01 appears when it should be non-significant.

JH: Thank you for carefully reading the manuscript and capturing the unpleasant error, it was, of course, insignificant (p=0.85), we should have noticed it.

Please review the statistical study thoroughly, in order to make the manuscript more rigorous, and take this aspect into account in case the discussion of the results is affected

JH: We checked the results and corrected several small errors coming from rounding and correction of several misprints when inputing data from primary protocols. The results of stat. analysis are ok.

Round 2

Reviewer 1 Report

Comments and Suggestions for Authors

Thank you for revising the manuscript and the paper has been improved in a major way. However, minor edition, including the use of ratios and percentage units in the text is recommended.

Comments on the Quality of English Language

Minor edition, including the use of ratios and percentage units in the text is recommended. 

Author Response

Dear reviewer,

thank you again for your valuable time devoted to an unusually careful and detailed study of our manuscript. Your comments and catching unnecessary mistakes have improved the quality of the article and will help us avoid repeating some mistakes in the future.

We have fixed errors (not removing the %) sign that occurred with some ratio values by replacing the percentage with a decimal expression. In some cases, where we thought it appropriate, we kept the ratio in percentages.

Reviewer 2 Report

Comments and Suggestions for Authors

Dear authors, I am very satisfied with your answers to my remarks. According to this, my opinion is that the manuscript should be accepted and published in its current form.

Author Response

Dear reviewer,

thank you again for your valuable time devoted to carefully studying our manuscript. Your comments and catching unnecessary mistakes have improved the quality of the article and will help us avoid some mistakes in the future.